# ExpressiveSinger: Multilingual and Multi-Style Score-based Singing Voice Synthesis with Expressive Performance Control

## ABSTRACT

Singing Voice Synthesis (SVS) has significantly advanced with deep generative models, achieving high audio quality but still struggling with musicality, mainly due to the lack of performance control over timing, dynamics, and pitch, which are essential for music expression. Additionally, integrating data and supporting diverse languages and styles in SVS remain challenging. To tackle these issues, this paper presents *ExpressiveSinger*, an SVS framework that leverages a cascade of diffusion models to generate realistic singing across multiple languages, styles, and techniques from scores and lyrics. Our approach begins with consolidating, cleaning, annotating, and processing public singing datasets, developing a multilingual phoneme set, and incorporating different musical styles and techniques. We then design methods for generating expressive performance control signals including phoneme timing, F0 curves, and amplitude envelopes, which enhance musicality and model consistency, introduce more controllability, and reduce data requirements. Finally, we generate mel-spectrograms and audio from performance control signals with style guidance and singer timbre embedding. Our models also enable trained singers to sing in new languages and styles. Several listening tests reveal both musicality and controllability of our generated singing compared with existing works and human singing. We release the data for future research. Demo: https://expressivesinger.github.io/ExpressiveSinger.

## CCS CONCEPTS

• **Applied computing** → **Sound and music computing**.

## KEYWORDS

Singing Voice Synthesis, Expressive Performance Control, Singing Style, Diffusion Model

## 1 INTRODUCTION

The Singing Voice Synthesis (SVS) task involves computer models automatically generating singing audio given symbolic music scores with lyrics. It has been a long-standing area of research since the 1930s [10, 21, 37, 40], with various applications in music production, entertainment, and education. Recent progress in deep learning has demonstrated remarkable potential in generating audio [26, 27, 30, 35, 54], with impressive results in voice modeling for Text-To-Speech (TTS) Synthesis [2, 38, 47, 48] and SVS

Permission to make digital or hard copies of all or part of this work for personal or classroom use is granted without fee provided that copies are not made or distributed for profit or commercial advantage and that copies bear this notice and the full citation on the first page. Copyrights for components of this work owned by others than the author(s) must be honored. Abstracting with credit is permitted. To copy otherwise, or republish, to post on servers or to redistribute to lists, requires prior specific permission and/or a fee. Request permissions from permissions@acm.org.

*ACM MM, 2024, Melbourne, Australia*

© 2024 Copyright held by the owner/author(s). Publication rights licensed to ACM.
ACM ISBN 978-x-xxxx-xxxx-x/YY/MM
https://doi.org/10.1145/nnnnnnn.nnnnnnn

[17, 31, 32, 56]. However, compared to the rapid advancements in TTS, SVS continues to face significant challenges.

One of the primary challenges in current SVS is insufficient musicality. Synthesized singing often suffers from issues such as being out of tune, unnatural techniques, and poor dynamics, which are closely related to *performance control* from a musical perspective. Music fundamentally relies on performance, where musicians interpret scores with their personal styles and emotions. Performance control encompasses critical music elements often missing in symbolic scores, such as performance timing, dynamics, pitch contour, timbre control, and playing techniques, which are key to making generated music sound natural. This is comparable to a masterful violin that, regardless of its high quality, will sound vastly different in the hands of a professional compared to a novice, underscoring the importance of skilled performance control.

Most deep learning-based SVS systems are directly adapted from TTS models, where the synthesizers are designed for speech and lack the capability to address challenges in music performance. For instance, singing encompasses a broader pitch frequency range and displays greater timbre texture diversity, including breathiness, chest voice, and head voice, compared to speech. These aspects are typically absent in speech synthesis. Most importantly, SVS generates singing from symbolic scores instead of just text (lyrics), requiring a significantly higher level of expressive performance control mastery than speech. Many SVS systems borrowed from TTS models cannot process actual sheet score input and instead rely on ground truth performance control signals, such as performance MIDI, phonetic timing, pitch and loudness curves, as input conditions. Such systems that take advantage of real singing data should not be considered full-stack score-based SVS.

Given the high audio quality of generated human voices, we shift our focus towards musicality and expressive performance control, which we deem as the primary bottleneck in SVS. We design a cascade of diffusion models [14, 42] to generate the three most critical control signals in expressive performance: performance timing, pitch (Fundamental Frequency (F0)) contour, and dynamics (loudness curve). These control signals are generated according to not only score and lyrics, but also music genre style, singing technique, and singer identity. They serve as the foundation for generating the final singing audio and controlling the musicality. We choose diffusion models and related extensions as the model architecture due to their strong performance in modeling continuous representations, such as images and audio.

Besides expressive performance control, the current SVS systems face two other significant challenges. On the one hand, there has been a consistent lack of high-quality public singing datasets with annotations. Unlike the extensive datasets available for speech, singing datasets are typically small, with considerable variation in recording environments, data quality, singer proficiency, and

annotations, making integration challenging. On the other hand, existing SVS systems still struggle to incorporate multiple languages, diverse musical styles, different datasets, and inconsistent acoustic environments simultaneously.

To address the data issue and expand the capacities of SVS systems, we first clean up and consolidate publicly available singing datasets, adding necessary annotations, refining approaches to extracting acoustic features for singing, and categorizing datasets to optimize their utilization across different stages of our SVS system. Second, we develop a multilingual phoneme set, merging phoneme sets from different datasets to enable multilingual generation. Moreover, we introduce a range of musical styles and techniques into the system and employ speaker/singer embedding instead of the traditional singer ID to utilize the training data efficiently.

In all, we aim to explore expressive performance control in singing and generate multilingual and multi-style singing voices with both high musicality and audio quality. Our solution, *ExpressiveSinger*, involves three modules: expressive performance control signal generation, mel-spectrogram generation, and audio waveform generation. Evaluations include multiple subjective assessments to demonstrate the effectiveness of our system compared to previous works and human singing.

We summarize our contributions as follows: (1) Introduce a comprehensive SVS system that generates expressive and realistic singing from scores and lyrics with multiple languages, styles, techniques, and singers from symbolic music scores with lyrics. (2) Design a set of methods for generating expressive performance controls, which not only improve musicality and consistency of the synthesized singing, but also allow more controllability and reduce data requirements. (3) Combine, clean, annotate, and process various public singing datasets, and release the integrated data for future research. (4) Demonstrate the zero-shot capabilities of our models by enabling singers from the training data to sing in languages and styles they have not previously attempted.

## 2 RELATED WORK

### 2.1 Singing Voice Synthesis

Voice modeling can be traced back to Bell Labs [10], and vocal synthesizers like VOSIM [21] and FOF [40] have been widely used in the industry. Recently, deep learning approaches in audio generation and TTS, starting from Wavenet [35], deep acoustic models [38] and neural vocoders [24–26, 28], to audio codecs [27, 54], have also become the mainstream in SVS. Score-based SVS systems [12, 32] process symbolic scores and lyrics as input, while other SVS systems [31, 39] input lyrics with some ground-truth performance controls, such as performance timing for each word or phoneme, pitch and loudness curves. The significant difference between performance MIDI and the actual sheet music score is often overlooked and misunderstood in SVS research. Performance MIDI contains expressive performance timings rather than the regular beat-based note durations in scores. Also, pitches in the performance MIDI including techniques like grace notes, ornaments, and glissandos, may differ from those in the score. Using performance MIDI for SVS while claiming it to be score-based is misleading. In this paper, our system takes scores instead of performance MIDI as input.

A widely used architecture in TTS and SVS is a two-step synthesis process: an *acoustic model* that converts the input to an acoustic representation, and a *vocoder* that synthesizes the final audio output from this representation. The acoustic representation can be standard formats like spectrograms or mel-spectrograms, or other pre-trained representations and templates such as audio Encodec codes [17] and DDSP harmonic representations [56]. Common practices for *acoustic models* include transformer-based models [16, 31, 38], as well as WaveNet and FFT (Fast Fourier Transform)-based methods [32, 57]. These models are often effectively paired with GANs [57] or diffusion processes [31]. For *vocoders*, deep learning [24, 25, 53] has made significant progress. For instance, BigVGAN [28] integrates periodic activation functions and anti-aliased multi-periodicity composition, yielding high-fidelity speech and music synthesis. Diffwave [25], leveraging a Denoising Diffusion Probabilistic Model (DDPM) [14] with a WaveNet backbone, offers ease of training and high-quality. To improve pitch sensitivity in singing, some studies [57] have incorporated quantized F0 curves as additional input for the vocoder. This paper uses BigVGAN as the vocoder with both inputs of the generated mel-spectrograms and F0 curves. We also take inspiration from the architecture of Diffwave for the acoustic model and performance control generation.

Current deep-learning-based SVS systems struggle to incorporate multiple languages, diverse genre styles, various singing techniques, and inconsistent acoustic environments from different datasets. First, due to data scarcity and non-unified representations, most SVS models can only handle one language at a time, with an extremely unbalanced focus on Chinese Mandarin [12, 32, 56, 57]. Second, existing models are limited to one music genre, predominantly popular music, with only a small portion focusing on other genres [22, 58]. Third, they cannot generate singing based on singing technique prompts, such as lip trill and vibrato [50]. In addition, most models employ no more than three datasets, and it remains unknown how to handle the varying acoustic environments of different singing datasets while generating consistent and high-quality audio. Finally, models have not explored enabling singers to sing in languages or styles that they have never sung by themselves in the training set. In this paper, we introduce solutions to all these challenges.

Some SVS systems have explored generating multiple singer voices [39, 57]. However, they use a numerical representation of Singer ID to differentiate between singers in the training set, which hinders the model's extensibility and needs retraining the ID encoder when adding new singers. Moreover, it is difficult for the model to share and generalize data across different singers, requiring a notable amount of training data for each singer. Here we adopt the Resemblyzer speaker embedding [46] instead of ID in the acoustic model, drawing inspiration from TTS and speech conversion models [2, 44, 47].

### 2.2 Expressive Performance Control

Expressive performance controls can be mainly categorized into timing, pitch, dynamics, and timbre control [8, 29], with significant effects on music perception and expression [4, 20]. Besides a few attempts at emotion expression control [23], most deep-learning-based SVS models either implicitly model expressive performance

controls or include these modules in an end-to-end training fashion [31, 32, 56, 57]. This approach leads to various issues. First, the results are inconsistent and lack musicality, often exhibiting issues like out-of-tune, erratic timing and volume. Second, it is hard to precisely control the corresponding performance attributes for different music styles and singers, such as swing timing and pitch bends in jazz. Finally, it requires a large amount of high-quality training data, and cannot fully utilize all types of available data. For instance, data with low audio quality may still contain professional performance data, making it unsuitable for training the acoustic model or vocoder but ideal for the performance control model.

Before the advent of deep learning, rule-based [1] and traditional machine learning approaches [36, 43] made notable progress in modeling expressive performance control for singing and instruments. First, performance timing, different from score timings, often involves rhythm and tempo variations. Most studies modeling timing [41, 52] focus on piano, and use machine learning to model deviations between note onsets in performance timing and the original score timing. Second, continuous pitch variation curves during performance are typically analyzed using Fundamental Frequency (F0), and closely tied to playing techniques like vibrato, glissando, and ornaments [15, 19]. Moreover, dynamics involve the loudness and softness of notes, and researchers often use amplitude envelopes (curves) extracted from audio performance to represent dynamics control based on music context [5, 6, 9, 15, 51]. This paper is the first work that utilizes the diffusion process with deep learning architectures to explicitly model expressive performance control from scores and styles, including timing onset deviations, F0 curves, and amplitude envelopes. We leave timbre control to the model implicitly due to lacking timbre recognition algorithms and annotations.

## 3 DATA PREPARATION

### 3.1 Dataset Integration and Categorization

First, we utilize the publicly available singing datasets that contain solo singing with minimal noise, despite varying acoustic environments and sound quality, including SingStyle111 [7], Opencpop [49], M4Singer [55], Children Song Dataset (CSD) [3], VocalSet [11, 50], PopCS [31], and OpenSinger [16].

Next, we have dedicated substantial effort to data cleanup and correction, adding essential annotations. For instance, we manually correct the frequently incorrect pitch annotations in Opencpop, which were often off or up by an octave. We also manually annotate the tempo for half of the songs in M4Singer, as all the provided performance MIDI files used a uniform tempo of 120 bpm. In CSD dataset, we segment each song's complete audio and performance MIDI into shorter phrases based on the lyrics' syntax, for model batch training. We complete the missing lyrics, phonemes, and their corresponding audio position and duration annotations for the songs in VocalSet. For datasets like M4Singer, CSD, and Opencpop that have performance MIDI but no scores, we quantize the performance MIDI to a minimum grid of 32nd notes based on each song's tempo, creating quantized scores aligned with lyrics phonemes and words[1] for each phrase.

---

[1]In Chinese Mandarin and Korean, "word" refers to character.

Finally, we categorize the datasets and map them to three modules of our system: expressive performance control, acoustic model, and vocoder. The vocoder training use all seven datasets (in total 118.67 hours, 121 singers); the acoustic model and expressive performance control (F0 and amplitude curve) training use SingStyle111, Opencpop, M4Singer, CSD, and VocalSet (in total 62.78 hours, 50 singers) due to their better sound quality and phoneme/word duration annotations. The performance timing model only utilizes SingStyle111, Opencpop, and part of M4Singer data (in total 30.6 hours, 22 singers), as score input and phoneme duration annotations are required. After integration, all data were resampled to both 44.1 kHz and 22.05 kHz audio waveforms and segmented into short phrases ranging from 2 to 20 seconds in length.

### 3.2 Data Representation

We create a multilingual phoneme set covering English, Chinese, Italian, and Korean. We merge phoneme sets from different datasets, such as the International Phonetic Alphabet (IPA), Advanced Research Projects Agency (ARPA) Phonetic Set, CMU Pronouncing Dictionary, Mandarin Pinyin, etc. Converting between phoneme sets is not straightforward; for example, one Mandarin Pinyin phoneme may correspond to multiple IPA phonemes, and splitting Pinyin into IPA would result in losing the original phoneme duration annotations. To address this, we manually merge phonemes with the same pronunciation from different sets while retaining the distinct phonemes. Even though this is not a standard phoneme set, we have made it flexible to ingest different languages. The final set contains 95 phonemes indexed by number, including three non-phonetic sounds: AP (aspirate), SP (silence), and NS (noise and other unknown sounds).

We design style and technique tokens for style control. The six style genres are pop, children, Western opera, traditional Chinese folk, jazz, and Teresa (singer *Teresa Teng*). Each song can have multiple styles, represented by a six-dimensional binary vector. For instance, a song combining jazz and traditional Chinese folk would have 1s in those dimensions and 0s elsewhere. Similarly, musical theater songs typically blend pop and Western opera styles. For techniques, we adapt the 16 opera techniques from VocalSet (e.g., lip trill, trillo, belt) plus a "normal" indicating no specific technique, using one-hot encoding. For other datasets besides VocalSet, we only use "vibrato" and "normal" labels. Initially, we included more detailed style genres (e.g., rock, country, musical) and four emotion types (normal, happy, lyrical, and exaggerated), but found that overly detailed and inaccurate classifications confused the model, especially with a relatively small data scale. We retain these detailed labels in the released data for future research.

Our acoustic model uses data with manually annotated phoneme-audio alignment, including each phoneme's start time and duration in seconds. For datasets with only word-level alignment, a rule-based algorithm (detailed in Section 4.2.1) splits the word-level duration into phoneme-level durations. Performance MIDI information is added to the phoneme alignment, including each phoneme's corresponding note pitch (MIDI pitch numbers 1-127) and word boundaries (also performance MIDI note boundaries).

The score representation is a list of notes, each containing a MIDI pitch number (0 means rest note) and a duration in 32nd note

units, which may differ in length from the phonemes. For instance, {(60, 4), (62, 4), (64, 8), (0, 16)} represents two eighth notes (C4 and D4) followed by a quarter note (E4) and a half note of rest. We also provide the alignment between notes and words in the score, allowing for one-to-many and many-to-one correspondences.

Lastly, singer information in the acoustic model is represented by a 256-dimensional speaker embedding vector [46]. However, we continue to use singer ID in expressive performance control models since performance control signals should be disentangled from voice timbre. The song's original dataset is described as a one-hot encoded ID vector.

## 3.3 Acoustic Feature Processing

We extract mel-spectrograms from 22.05 kHz audio using the Short-Time Fourier Transform (STFT) with a window size of 1024, FFT size of 1024, hop size of 256, and bin size of 80. To obtain amplitude envelopes (loudness), we calculate the root-mean-square (RMS) amplitude values from audio using the same STFT settings and convert them to decibels. A moving average window of frame size 30 is applied to smooth the amplitude curve.

To analyze accurate F0 curves from singing, we combine pYIN [33], PENN [34], and Parselmouth [18]. pYIN and Parselmouth are used to determine unvoiced parts (breaths, silence, consonants). For voiced parts, we choose the PENN result if it differs from Parselmouth or pYIN by no more than one semitone; otherwise, we select the Parselmouth result. Smoothing is applied to address common octave errors in high-frequency components.

## 4 METHOD

For each data segment (typically a musical phrase), our SVS system, *ExpressiveSinger*, takes score, lyrics, style tokens, and singer information as input and generates expressive and realistic singing in the audio waveform. As shown in Figure 1, the pipeline involves three main modules: (1) three expressive performance control models that generate three types of control signals: performance timing at phoneme level, amplitude envelopes, and F0 curves; (2) an acoustic model that generates the mel-spectrograms conditioning on performance control signals; (3) a vocoder to generate the waveform from mel-spectrograms and F0 curves.

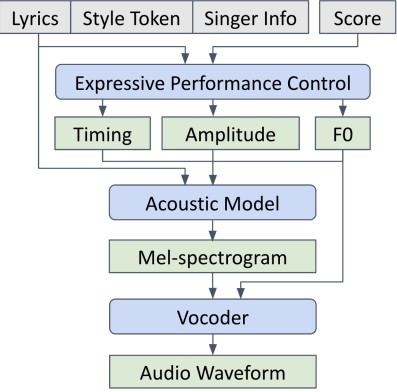

**Figure 1: Pipeline of ExpressiveSinger.**

## 4.1 Model Architecture

The three expressive control models and the acoustic model share a similar architecture inspired by Diffwave [25] and WaveNet [35], but with notable differences. We employ diffusion-based training and inference. In the training stage, diffusion process is defined as a Markov chain gradually converting real data $x_0$ to whitened latent variable $x_T$, with Gaussian transitions parameterized by a decreasing sequence $\alpha_{1:T} \in (0, 1]^T$ in Eq.(1). We can also express $x_t$ as a linear combination of $x_{t-1}$ and a noise variable $\epsilon$ in Eq.(2).

$$q(x_{1:T}|x_0) = \prod_{t=1}^{T} q(x_t|x_{t-1}) = \prod_{t=1}^{T} \mathcal{N}(\sqrt{\frac{\alpha_t}{\alpha_{t-1}}}x_{t-1}, (1 - \frac{\alpha_t}{\alpha_{t-1}})I),$$
$$(1)$$
$$x_t = \sqrt{\alpha_t}x_{t-1} + \sqrt{1 - \alpha_t}\epsilon, \quad \epsilon \sim \mathcal{N}(0, I) \qquad (2)$$

In the reverse process, instead of DDPM, we utilize Denoising Diffusion Implicit Models (DDIM) [42] which allows non-Markovian (implicit) generation and accelerates the inference. We select a subsequence $\tau$ out of $[1, \cdots, T]$ with length $S$ and $\tau_S = T$, then reverse process denoising from $x_T$ to $x_0$ parameterized by $\theta$ is:

$$p_\theta(x_{0:T}) = p_\theta(x_T) \prod_{i=1}^{S} p_\theta^{(\tau_i)}(x_{\tau_i-1}|x_{\tau_i}) \times \prod_{t \in \bar{\tau}} p_\theta^{(t)}(x_0|x_t),$$
$$\text{where} \quad p_\theta(x_T) = \mathcal{N}(0, I), \quad \bar{\tau} = [1, \cdots, T] \backslash \tau, \qquad (3)$$

Since $p_\theta^{(t)}(x_0|x_t)$ only involves in the variational objective, we are able to speedup sampling with fewer steps S rather than T. Similarly, we can express the closed form equation in DDIM as:

$$x_{\tau_{i-1}} = \sqrt{\alpha_{\tau_{i-1}}}\left(\frac{x_{\tau_i} - \sqrt{1 - \alpha_{\tau_i}}\epsilon_\theta(x_{\tau_i})}{\sqrt{\alpha_{\tau_i}}}\right) + \sqrt{1 - \alpha_{\tau_{i-1}}}\epsilon_\theta(x_{\tau_i}), \quad (4)$$

Different from DDPM, in DDIM, the forward process becomes deterministic given $x_{t-1}$ and $x_0$, except for $t = 1$; and the generation also becomes a fixed procedure from latent variables.

Figure 2 illustrates the model architecture for predicting noise $\epsilon_\theta$ at each diffusion step $t$. The input $x_t$ varies depending on the model, ranging from control signals to mel-spectrograms. Additional inputs include the diffusion step, $t$, and contextual conditions like singer information, lyrics, and style tokens, which also vary by model. These inputs are processed through encoders to enhance embeddings before entering residual layers. Inspired by Wavenet [35], each residual layer incorporates a bi-directional dilated convolution and a gated-tanh activation function. The output from each layer is routed in two directions: to the final output through skip connections aggregating with other layers' outputs, and to the next residual layer as the subsequent embedded input, $x_{t+1}$. The diffusion step encoder uses the Diffwave[25] design, followed by two fully connected layers with swish activation. All convolution layers are initialized using Kaiming normal distribution [13], while the last layer before the final output utilizes zero initialization.

The condition context encoder also shares a similar architecture (Figure 3) but with varied inputs (detailed context input items for each model in supplementary materials). Each context item is processed through distinct embedding architectures before being concatenated and projected together. The lyric phonemes, consistently included in the condition context, are encoded using six Transformer [45] encoder layers with multi-head attention, followed by a

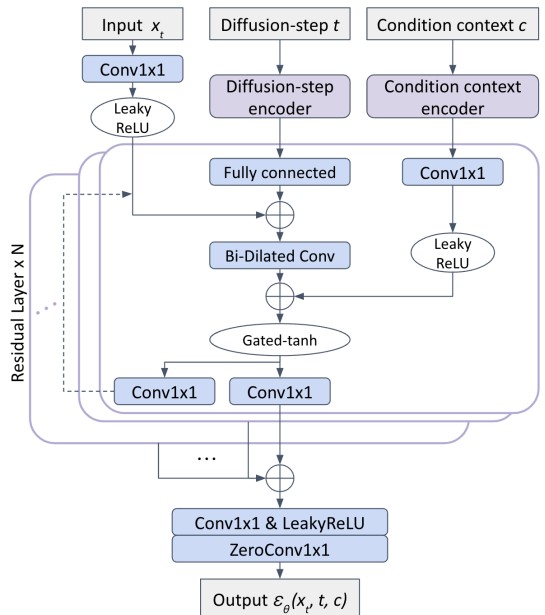

**Figure 2: Architecture for three performance control models and the acoustic model.**

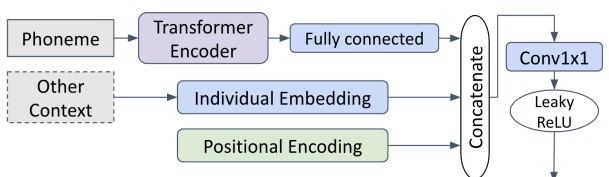

**Figure 3: Architecture for condition context encoder.**

fully connected layer. Given the model's non-autoregressive nature, we integrate necessary positional encodings to capture sequential dependencies, such as frame/beat position within each segment phrase, each phoneme, and each score note. Details on embedding layers for various condition contexts are provided in supplementary materials.

## 4.2 Expressive Performance Control

In this module, we generate expressive performance timing at the phoneme level using score, lyrics, singer, and style tokens as input. The generated timing is then used in models to generate F0 curves and amplitude envelopes.

*4.2.1 Expressive Timing.* Our expressive timing model inputs the score along with word-level aligned lyrics, generating performance timing onsets for each phoneme. Style tokens and singer information are also included in the input condition context to incorporate personalized style control. Notice here we focus on modeling onsets, omitting durations and offsets, as rests are treated the same as regular notes. This implies a note's offset is the same as the subsequent note's onset, allowing a sequence of note onsets to implicitly define durations and offsets.

As illustrated in Figure 4, the generation process contains two stages. First, a rule-based algorithm splits the score-word timings counted in beats, into each phoneme's timing in seconds, without changing word boundary timings. It primarily accounts for the differences between vowel and consonant phonemes, detailed in supplementary materials. In the second stage, instead of direct modeling onset timing, we employ the diffusion model in Section 4.1 to generate the onset deviations between the rule-based score phoneme timings and the final phoneme timing outputs, which simplifies training under the assumption of a diffusion Gaussian distribution. In particular, model input $x$ refers to onset deviations $[\sigma(1), \cdots, \sigma(n)]$, where $\sigma(i) = perform\_onsets(i) - score\_onsets(i)$, $i \in [1, n]$, $n$ = length of phonemes in the data segment. $score\_onsets(i)$ is the rule-based phoneme onsets in the first step, and $perform\_onsets(i)$ is the final output of this expressive timing model.

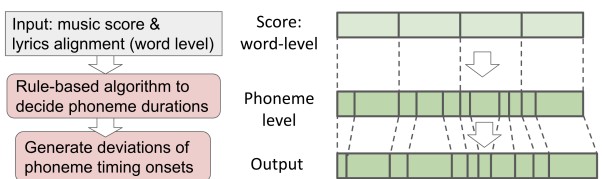

**Figure 4: Pipeline for generating expressive timing.**

*4.2.2 F0 Curves and Amplitude Envelopes.* The generation of F0 curves and amplitude envelopes is facilitated by two distinct yet structurally identical models, as described in Section 4.1. Their model condition context inputs are different from the timing model by (1) substituting score timing with the generated performance timing and (2) utilizing frame-wise positional encoding. To ensure compatibility with subsequent mel-spectrogram and audio waveform synthesis, we generate F0 curves and amplitude envelopes with the same lengths corresponding to the frame lengths of targeted mel-spectrograms. Consequently, the condition context for each phoneme needs to be expanded to mel-spectrogram frame length based on the generated phoneme timing. For the F0 model, input $x$ is defined as $[F0(1), \cdots, F0(m)]$, where $m$ represents the frame length of the target mel-spectrogram. Similarly, for the amplitude model, $x = [amp(1), \cdots, amp(m)]$. Prior to training, F0 data is linearly transformed to the range $[-1, 1]$ and amplitude data is normalization to $\mathcal{N}(0, I)$ in order to conform to approximate Gaussian distribution in the diffusion process, and are denormalized during the sampling.

## 4.3 Acoustic Model

The acoustic model adheres to the same architecture in Section 4.1, incorporating lyrics, style tokens, and singer timbre along with performance control signals (phoneme timing, F0 curves, and amplitude envelopes) generated by the previous module. This model excludes score information and purely depends on expressive performance control. Moreover, it notably benefits from the inclusion of quantized F0 curves, where F0 values in Hz are quantized into 256 discrete bins. Furthermore, the model utilizes singer embeddings,

which capture the unique timbre of each singer's voice, rather than singer IDs, enhancing the model's ability to generalize across different singers. Positional encodings remain consistent with those used in the F0 and amplitude generation models. Specifically, the input $x$ is represented as a 2D mel-spectrogram with 80 bins. Each bin of the mel-spectrograms is min-max normalized independently to the range $[-1, 1]$ before training.

## 4.4 Vocoder

We adapt BigVGAN [28] as the vocoder to synthesize the final audio waveform from mel-spectrograms. We incorporate quantized F0 curves as an additional conditioning input, using the same quantization methods as in the acoustic model. Furthermore, we modify BigVGAN's F0 frequency range to 11kHz to include high-frequency components present in singing.

## 5 EXPERIMENT AND EVALUATION

### 5.1 Experiment Settings

The data used in our experiments are detailed in Section 3. We selected 89 minutes of test data from different dataset sources in our collection, ensuring proportional representation and excluding any songs from the training set. The experiments were conducted using audio with a sample rate of 22.05 kHz.

In the diffusion process, the acoustic model has 1,000 diffusion steps; the F0 curve and amplitude envelope models both have 500 steps; and the performance timing model has 200 steps. The noise schedule, $\beta$, linearly increased from 0.0001 to 0.02. Diffusion step embeddings featured 128, 512, and 512 channels for the input, middle, and final layers, respectively. Each model incorporated 50 residual layers with a channel size of 256 and a dilated convolution cycle of 5. The acoustic model has trained over 2 million iterations with a batch size of 4 per GPU on an 8x A100 GPU cluster machine (distributed training involved). The F0, amplitude, and timing models were trained with 3 million iterations with the same batch size 4 on a 4x V100 GPU cluster for each model. Vocoder follows the same model and training setting with BigVGAN [28] but with additional quantized F0 as input condition.

We conduct multiple subjective listening tests and opt not to use the objective evaluation metrics typically employed in some TTS models, as they do not align well with highly musical singing voices. For example, F0 Root Mean Square Error (RMSE) measures discrepancies between the ground truth and predicted F0 contours. However, given the multiple expressive possibilities within a single musical score, including techniques like vibrato, glissando, and ornamentation, these can lead to distinctively different but equally pleasing F0 curves. Although a high F0-RMSE is considered unfavorable in speech, it may indicate a more expressive and musical performance in singing, especially when we involve more music styles and techniques in the model. Furthermore, RMSE for timing suffers from a similar multi-mode nature to F0-RMSE, with even more unreliability because singing lacks precise phoneme alignment algorithms like those in speech. Most musical timing data are manually marked with significant inconsistency and ambiguity, reducing accuracy. All these make RMSE comparison results in singing not informative.

## 5.2 Subjective Evaluation

### 5.2.1 Comparison With Existing Works and Human Singing.
We compare our model against the current state-of-the-art models, VISinger2 and DiffSinger, as well as with ground truth (GT) human singing. Notably, our model can synthesize different styles, techniques, languages, and multiple singers, which are not present in VISinger2 and DiffSinger. For DiffSinger, we utilize its end-to-end model available in their GitHub repository, where the F0 is generated implicitly, unlike the version described in their paper that requires GT F0 input. Additionally, both models use performance MIDI timing input from the OpenCpop dataset, rather than an actual quantized score timing. Consequently, our comparison has to be confined to the Chinese pop data. Our model's architecture remained unchanged but was trained only on the OpenCpop dataset for comparison. We use quantized OpenCpop scores as inputs, posing a bigger challenge to derive performance timing from score timing compared to the other two models.

The results are evaluated using the Mean Opinion Score (MOS). We conduct listening tests and collect valid feedback from 118 users, with over 90% being native Chinese speakers or those with a Chinese Mandarin learning background. Each evaluated five sets of results, with each set containing three audio samples of the same musical phrase: one from our model and two randomly selected from GT, VISinger2, and DiffSinger outputs. The five sets of phrases were randomly chosen from 206 OpenCpop test phrases, and the order within each set was also randomized.

The results are shown in Table 1. Our model significantly surpasses VISinger2 and DiffSinger in MOS and is close to the ground truth human singing. We note that in fast-tempo phrases, the difference between our model and the others was relatively small. However, in phrases that contain longer and more lyrical notes, our model demonstrates better musicality compared to the other models which frequently show erratic F0 control. This discrepancy likely arises from two reasons: (1) the primary challenge in generating fast-tempo singing is timing control, and both VISinger2 and DiffSinger use GT performance MIDI timing as input, instead of quantized score timing, thus bypassing this issue; (2) fast-tempo singing, being closer to speech, have lower musical demands for F0 and amplitude technique modeling, making them easier to handle. In contrast, long notes often require precise F0 control like vibrato, where our model consistently outperforms the others. Figure 5 displays the F0 control for a phrase having a long note. In the note circled by a black dashed line, both GT and our model demonstrate stable vibrato and accurate pitch. Conversely, VISinger2 and DiffSinger are notably out-of-tune with lower pitch and exhibit highly unstable vibrato, resulting in unnatural and unmusical singing.

### 5.2.2 Quality Of Style, Language, Techniques.
We evaluate our system's ability to generate different styles, languages, and techniques through a listening test. Here, we used the Comparison Mean Opinion Score (MOS) to compare the ground truth singing with the synthesized singing from our system. The experiments are conducted entirely based on the data configurations described in Section 3. We test all four languages and six style genres. For singing techniques, we select five representative ones for evaluation, including lip trill, trill, vibrato, trillo, and breathy singing. According to the results

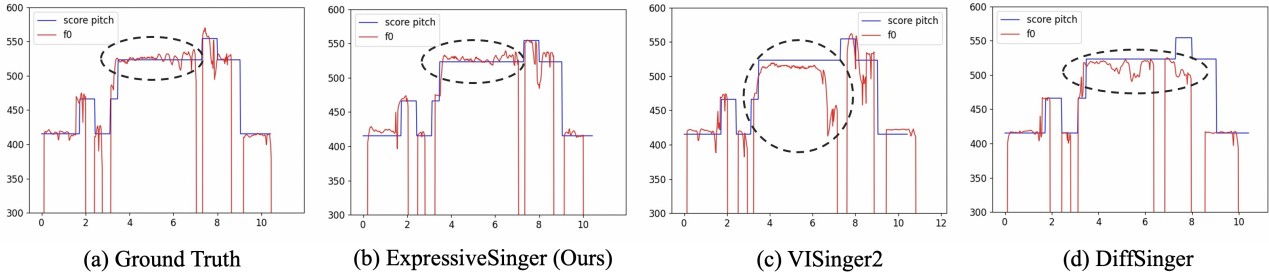

|  |  |  |  |
|:---:|:---:|:---:|:---:|
| (a) Ground Truth | (b) ExpressiveSinger (Ours) | (c) VISinger2 | (d) DiffSinger |

**Figure 5: Comparison of F0 curves from the generated singing of different models and human ground truth. The Y-axis represents the frequency value in Hz. The X-axis is the time frame, each unit is half-second. Red lines denote F0 curves,** 0 **value means unvoiced parts like consonants and breath events. Blue lines are score note pitches in frequency.**

**Table 1: Synthesized quality comparison among our model, GT human singing, and existing works. MOS score with 95% confidence interval.**

| System | Sample Rate | MOS |
|---|---|---|
| GT | 22kHz | 4.145 ± 0.097 |
| VISinger2 (MIDI timing) | 22kHz | 3.499 ± 0.115 |
| DiffSinger (MIDI timing) | 24kHz | 3.209 ± 0.129 |
| ExpressiveSinger (Ours, Score) | 22kHz | **3.956 ± 0.085** |

(see supplementary materials for details), our model achieves realistic generated singing very close to human singing across different languages and styles. However, opera singing quality is slightly lower than other styles, and the techniques of trillo and breathy sound are less successful compared to the other three.

*5.2.3 Ablation Study For Expressive Performance Control.* To verify the effectiveness of expressive performance control in our system, we conduct an ablation study. We modify the model by removing the expressive performance control module from the pipeline shown in Figure ??. Instead, inputs such as score, lyrics, and style tokens are fed directly into the acoustic model, using the same diffusion process and model architecture for training. We evenly divide phoneme durations within each word duration to provide the score timing input to the modified system. The Comparative Mean Opinion Score (CMOS) results from the listening tests, shown in Table ??, indicate a significant decline in model quality without explicit expressive performance control. This decline is not observed in erratic dynamics, frequent pitch instability, and inconsistent timing, as well as in a more robotic timbre with artifacts, demonstrating the crucial role of expressive performance control in achieving both natural and musical singing.

**Table 2: Ablation study for expressive performance control.**

| System | CMOS |
|---|---|
| ExpressiveSinger with EPC | 0.000 |
| ExpressiveSinger w/o EPC | -1.379 |

*5.2.4 Zero-shot Synthesis Scenarios.* Finally, we evaluate our system's ability to control and switch between different styles, languages, and techniques, particularly under zero-shot scenarios where the training dataset's singers had not previously attempted these variations. For example, we questioned whether singers who had only performed in Chinese pop could, with the system's help, sing in English, Italian, or Korean, or attempt opera, while retaining their unique vocal timbre characteristics. Additionally, we investigated whether our design of a combined multilingual phoneme set and the replacement of singer ID with singer embedding enhanced performance.

We design four ablation situations for these zero-shot scenarios, noting that no ground truth singing is available for comparison. The first scenario includes generated segments where the singer had experience in the same language and style within the training data. The second is a zero-shot scenario where the singer had no prior exposure to the segment's language and style. The third one uses traditional singer ID instead of singer embeddings under zero-shot scenarios. The final situation is to use unmerged, directly concatenated phoneme sets from all datasets instead of the combined phoneme set.

Subjective evaluations are conducted using the Mean Opinion Score (MOS), with detailed findings presented in the supplementary materials. We find minimal quality differences between zero-shot and non-zero-shot scenarios for some styles and languages, like Chinese pop. However, opera and zero-shot performances in Italian and Korean are less satisfactory, likely due to limited representations of them in the training data, such as only one singer performing in Korean, exclusively in children's songs. Furthermore, our results indicate that using singer embeddings in the acoustic model under zero-shot conditions provided better quality than using traditional singer IDs. The implementation of a combined phoneme set also shows improvements in linguistic zero-shot scenarios.

## 6 CONCLUSION

ExpressiveSinger is a robust SVS system that processes scores with lyrics to generate expressive and realistic singing across multiple languages, styles, techniques, and singers. The key idea is to emphasize expressive performance control, including timing, pitch contour, and dynamics, significantly enhancing the musicality and

naturalness of the synthesized singing, as demonstrated in our experiments.

Our pipeline eschews an end-to-end approach in favor of a three-stage process that offers greater controllability, more efficient use of diverse types of training data, and reduced data requirements. The effectiveness of our system and architectural design is validated through subjective evaluations, illustrating our model's capability to generate new styles and languages previously unattempted by the singers in the training data.

We also devote a significant amount of effort to data cleaning, annotation, combination, and processing, addressing the data scarcity challenges inherent in SVS.

Looking forward, I aim to refine the modeling of expressive performance controls and incorporate additional control signals like explicit timbre control. I also plan to enhance the controllability of styles and techniques in zero-shot scenarios. Ultimately, I aspire to develop a model capable of generating singing without relying on existing training data, pushing the boundaries of what is possible with synthesized voices.

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
