# OpenReview forum: "ExpressiveSinger: Multilingual and Multi-Style Score-based Singing Voice Synthesis with Expressive Performance Control"
_acmmm.org/ACMMM/2024/Conference — MM2024 Poster_

### Official Review · Reviewer_ZeEh · 2024-05-23

**Rating:** 4
**Confidence:** 3

**Summary:**

The paper introduces ExpressiveSinger, a system for synthesizing singing voices in various languages, styles, techniques, and for different singers. It uses a publicly available dataset and a Diffusion-based model for training. Results indicate that ExpressiveSinger produces reasonably high-quality synthesized singing.

**Strengths:**

- The problem addressed in this paper is crucial to the field.
- The writing is clear and easy to understand.
- As mentioned in Section 3, the paper integrates a publicly available singing voice dataset, enhancing the efficiency of diverse training data use and reducing data requirements. This is a great approach, given the relatively small size of singing voice datasets compared to speech datasets.
- Additionally, the paper details data usage during model training, offering valuable insights for future research.

**Limitations:**

- I would prefer if the author listed all existing expression controls in singing voice synthesis (e.g., emotions, singer ID, vibrato, singing technique) and the corresponding work for each. You may refer to this paper "Expression Control in Singing Voice Synthesis". Additionally, please explain why you chose to focus on the specific control signals in your paper.
- One major issue with the paper is the lack of discussion on disentangling different expressive controls (singer ID, style, singing technique, etc.). Have you addressed this issue, and if so, how? I would like to hear more about your approach to this challenge.
- The evaluation of controllability is somewhat lacking, as it only assesses the CMOS of the system. To better evaluate the control signals' effectiveness, I recommend conducting an evaluation where respondents identify the correct control signals from the synthesized audios. You might refer to the evaluation methods used in "SinTechSVS: A Singing Technique Controllable Singing Voice Synthesis System".

BTW, I cannot access your demo link. Can you provide an alternative file for your demo?

**Suitability:**

3

---

### Official Review · Reviewer_cNGY · 2024-05-24

**Rating:** 4
**Confidence:** 3

**Summary:**

The paper proposes a method, ExpressiveSinger, for singing voice synthesis by controlling the performance of timing, pitch contour, and dynamics with a three-stage process. Subjective results indicate that ExpressiveSinger significantly enhances the musicality and naturalness of the synthesized singing.

**Strengths:**

1. The paper is well structured with a clear introduction, detailed related work, and sufficient experimental analysis.
2. The paper is well motivated. Singing voice synthesis with both high quality and musicality is a challenging task. Since previous studies suffer from weak control on the performance of timing, dynamics, and pitch, ExpressiveSinger is a logical and promising solution.
3. The proposed method is novel and the generated results are better than other models.

**Limitations:**

1. Grammar and Style: Check the paper for spelling mistakes, awkward phrasing, and grammar errors to ensure the writing is smooth and refined.
2. Formatting Issues: Several formatting issues are present throughout the document. For example,
- line 733: "… shown in **Figure ??**. Instead, inputs such as score …"
- line 739-740: "… from the listening tests, shown in **Table ??**, indicate a significant decline in model quality without explicit …"
3. Incompleteness of Experiment Evaluation: There is only subjective evaluation for different models. However, objective evaluation also plays a crucial role in assessing the quality of generated results. Including objective evaluations could significantly enhance the credibility and authenticity of the model.
4. The authors did not provide details about the source code and model release. This is an important aspect of open research, as such releases will enable the scientific community to reproduce the results and use them in future research efforts.

**Suitability:**

3

---

### Official Review · Reviewer_NZ2B · 2024-05-25

**Rating:** 6
**Confidence:** 3

**Summary:**

The abstract discusses ExpressiveSinger, an advanced Singing Voice Synthesis (SVS) framework using diffusion models to improve musicality by controlling timing, dynamics, and pitch. It addresses data integration and supports diverse languages and styles by consolidating public datasets, creating a multilingual phoneme set, and incorporating various musical techniques. Expressive performance control signals enhance musicality and consistency, leading to the generation of audio that allow trained singers to perform in new languages and styles.

**Strengths:**

- SVS system that, starting form scores and lyrics, produces expressive singing allowing multiple languages, styles, techniques, and singers.
- The three-stage process offers greater controllability, more efficient use of diverse types of training data, and reduced data requirements
- Convincing subjective evaluation of the system
- Data integration and release from various public dataset are available for future research

**Limitations:**

I have only minor remarks
- line 773: Figure ??
- line 779: Table ??
- line 690: “(MOS)” =>  “(CMOS)”
References
- ref. 4: “..” => “.”
- ref. 5: “Comouter” => “Computer”
- ref. 7, 29: “In in” => “In”
- ref. 23: “Processing” => “, vol. 31, pp. 2751-2764, 2023, doi: 10.1109/TASLP.2023.3294712.”

**Suitability:**

3

---

### Meta-Review · Area_Chair_v7aD · 2024-06-23

**Recommendation:** Accept (Poster)
**Confidence:** 3

**Metareview:**

+ The problem addressed in this paper is important to the field.
+ The writing is clear and easy to understand.
+ The paper integrates a publicly available singing voice dataset and presents a data-efficient training method.

-  A major problem with this paper is the lack of discussion on disentangling different expressive controls (singer ID, style, singing technique, etc.). This weakness puts the paper as a borderline paper which should only be accepted if there is space. There is no consensus among reviewers about this paper.